# Inactivation of *Acanthamoeba* Cysts in Suspension and on Contaminated Contact Lenses Using Non-Thermal Plasma

**DOI:** 10.3390/microorganisms9091879

**Published:** 2021-09-05

**Authors:** Tereza Měřínská, Vladimír Scholtz, Josef Khun, Jaroslav Julák, Eva Nohýnková

**Affiliations:** 1Department of Physics and Measurements, University of Chemistry and Technology, 166 28 Prague, Czech Republic; vladimir.scholtz@vscht.cz (V.S.); josef.khun@vscht.cz (J.K.); 2Institute of Immunology and Microbiology, 1st Faculty of Medicine, Charles University and General University Hospital in Prague, 128 00 Prague, Czech Republic; jaroslav.julak@lf1.cuni.cz (J.J.); enohy@lf1.cuni.cz (E.N.)

**Keywords:** *Acanthamoeba*, cysts, contact lenses, DC corona discharge, IR spectra, Raman spectra

## Abstract

Water suspensions of cysts of a pathogenic clinical isolate of *Acanthamoeba* sp. were prepared, and the cysts were inactivated either in suspension or placed on the surface of contact lenses by the non-thermal plasma produced by the DC corona transient spark discharge. The efficacy of this treatment was determined by cultivation and the presence of vegetative trophozoites indicating non-inactivated cysts. The negative discharge appeared to be more effective than the positive one. The complete inactivation occurred in water suspension after 40 min and on contaminated lenses after 50 min of plasma exposure. The properties of lenses seem to not be affected by plasma exposure; that is, their optical power, diameter, curvature, water content and infrared and Raman spectra remain unchanged.

## 1. Introduction

The genus *Acanthamoeba* represents free-living amoebae, distributed worldwide and isolated from nearly all environmental niches, e.g., soil, dust, air, seawater and fresh water and plants. Acanthamoebae are considered to be the most common free-living eukaryotic microorganisms. They have two stages in their life cycle: (1) the metabolically active, motile trophozoite stage and (2) the dormant cyst stage [1]. Under stressful conditions, such as starvation and environmental extremes, the trophozoite converts to the cyst, which can survive for years [2,3]. Under favorable conditions, the cyst transforms back to the trophozoite in a process called excystation. Cysts are also responsible for the extraordinary resistance of acanthamoebae to drugs and disinfectants [4] as well as for persistent and/or recrudescent infections: in infected tissues, they survive for months (for review, see [5]). Acanthamoebae, although free living, can, under certain conditions, cause a spectrum of human infections from rare but highly fatal infection of the brain known as granulomatous amoebic encephalitis, mainly affecting immunocompromised persons, to infection of the cornea known as *Acanthamoeba* keratitis (AK) [6]. The corneal infection is manifested by keratoconjunctivitis and stromal ulcers, which can lead to blindness and affects otherwise healthy individuals. The main cause of AK is wearing contaminated/insufficiently disinfected contact lenses (CLs), which serve as “a vector” transmitting acanthamoebae from the environment to the corneal surface. Poor hygiene of CLs, as well as the usage of certain CL solutions, is among the major risk factors for AK (for review, see [5,7]). Commercial lens care disinfectant solutions in use guarantee a wide spectrum of effectiveness against pathogenic microorganisms, including acanthamoebae, within 6 h of exposure, which is the usual recommended disinfection time for most of them. However, in vitro tests showed low efficacy of many CL multipurpose solutions against *Acanthamoeba* cysts even after 6 h [8,9,10]. Therefore, there are efforts to find alternative approaches to CL disinfection, e.g., new chemical formulas of multipurpose contact lens solutions [8,11], conjugation of CL solutions with nanoparticles [12] and disinfection of CL storage cases using microwave treatment [13]. An unusual attempt to inactivate acanthamoebae using simulated global solar irradiance was described by Heaselgrave et al. [14]. Another approach could be using cold atmospheric plasma [15].

Plasma, or the fourth state of matter, is a partially or fully ionized gas. Whereas high-temperature plasma reaches temperatures of thousands of kelvins, non-thermal plasma (NTP), also low-temperature plasma or cold plasma, occurs at ambient temperature. It consists of low-temperature ions and highly energetic free electrons. NTP may be easily obtained by various electric discharges; for a more detailed description of plasma sources, see, e.g., [16,17,18,19,20,21,22].

NTP is widely used in various fields, including biology and medicine, as summarized in numerous reviews, such as [23,24,25,26,27], and in the book [28]. It is useful mainly as a disinfection agent, but it also accelerates blood coagulation and improves wound healing; applications in dentistry [29] and cancer therapy [30,31] have also been described.

The chemical composition of NTP is rather complex, as described by Graves [32], Kelly and Turner [33], Sysolyatina et al. [34] and Liu et al. [35]. Various reactive oxygen and nitrogen species (RONS) as ions, radicals and stable or unstable electro-neutral molecules are present. The lifetimes of these species are mostly very short with typical half-periods of life from nanoseconds to a few seconds. The stable compounds formed are hydrogen peroxide, ozone and nitrogen oxides NOx. RONS arising from surrounding gases mediate the microbicidal activity of NTP. Besides the direct action of plasma, the plasma-activated water (PAW) is also active for many months after plasma exposure due to the presence of stable RONS [36,37,38].

The disinfection effects of NTP have been studied mainly on bacteria, but attempts to inactivate viruses [39,40,41] or fungi both in vitro and in vivo [42,43] were also made. In general, the sensitivity of various microbes to NTP differs substantially: while the inactivation of bacteria is complete within seconds to minutes, yeasts require exposure for several minutes and mold and bacterial spores several tens of minutes. Moreover, microorganisms embedded in biofilm are significantly more resistant to NTP than those in planktonic form and, therefore, require a longer exposure for inactivation [44,45]. The rate of inactivation also depends on the experimental exposure parameters, the arrangement of which can be very different.

Reports on the inactivation of parasites using NTP are rather rare. Hayes et al. [46] evaluated a pulsed gas plasma discharge system for its ability to inactivate oocytes, a resistant dormant stage of the protozoan enteroparasite *Cryptosporidium parvum*, causing waterborne zoonotic cryptosporidiosis in humans and animals. Rowan [47] summarized the broader context of water purification using pulsed gas plasma, including inactivation of *Cryptosporidium* oocysts. Several reports on the inactivation of larval stages of parasitic helminths have also been published. Wang et al. [48] described the inactivation of *Schistosoma japonicum* cercariae using the dielectric barrier discharge in the stream of carrier He, O_2_ or air. Hejzlarová et al. [49] inactivated *S. mansoni* cercariae and miracidia using NTP produced by three types of DC corona discharges.

The inactivation of *Acanthamoeba* has already been described by Heaselgrave et al. [15] using the unique system of ambient air plasma confined to the surface of a metallic mesh used as the ground electrode. This system resembles the dielectric barrier discharge arrangement, operating at 20 kHz. Trophozoites of *A. polyphaga* and *A. castellanii* were completely inactivated in 1 and 2 min, respectively. For the highly resistant cyst stage, complete inactivation (the log10 reduction in viability of 3.75–4.42) was achieved after 4 min of exposure.

In this study, we used the DC corona discharge as a plasma source to inactivate the *Acanthamoeba* cysts both in water suspension and on contact lenses immersed in water, mimicking disinfection practice. We also observed the effects of plasma on the physical properties of two types of commercial CLs.

## 2. Materials and Methods

### 2.1. Acanthamoeba Strain

The pathogenic isolate BM-18AK-OP of *Acanthamoeba* sp. was derived by scraping of the corneal ulcer in a patient with clinically suspected *Acanthamoeba* keratitis. The corneal sample was examined by a routine diagnostic cultivation method on a 1.5% non-nutrient agar plate covered with a heat-inactivated suspension of *Escherichia coli* (see below) at 25 °C (Schuster 2002). The *Acanthamoeba* isolate was then axenized using a liquid Bacto-Casitone-Serum (BCS) medium [50] supplemented with 500 IU/mL penicillin and 125 µg/mL amikacin at 25 °C. In both culture systems, the vegetative forms of acanthamoebae (trophozoites) spontaneously transformed to cysts within seven to ten days of cultivation. The isolate was genotyped as T4 based on the 18S rDNA [51]. Pathogenicity of the isolate was demonstrated by co-culture with a monolayer of Vero kidney cells as described earlier [52].

### 2.2. Stock Suspension of Acanthamoeba Cysts

The spontaneously formed cysts, i.e., cysts formed by the starvation of trophozoites in stationary cultures, were harvested from the BCS medium and used for the experiments. A stock cyst suspension was prepared by centrifugation at 800× *g* for 10 min at room temperature. The pellet was re-suspended in 10 mL of distilled water, the cysts were counted using Bürker hemocytometer counting chamber, centrifuged again as above, and the pellet was re-suspended in distilled water to give 10^6^ cysts per ml. The stock suspension was stored at 4 °C.

### 2.3. Stock Suspension of Escherichia coli

The strain of *Escherichia coli* (CNCTC 6859), from the Czech National Collection of Type Cultures, was employed in experiments. Before use, the lyophilized bacteria were suspended in distilled water and inoculated on 2% nutrient agar (Imuna Pharm, Šarišské Michaľany, Slovakia) for 24 h at 37 °C. The bacteria were harvested from the agar surface, suspended in 2 mL of water and inactivated for 40 min at 65 °C in a water bath. The inactivated suspension was stored at 4 °C until use.

### 2.4. Contact Lenses

Commercially available hydrogel and silicon hydrogel soft hydrophilic CLs were investigated. The silicon CL brand was Soflens (Bausch&Lomb, Laval, Canada), made of Hilafilcon B (containing poly-2-hydroxyethylmethacrylate, polyvinylpyrrolidone, allylmethacrylate and ethylenglycoldimethacrylate), swelling 59%. The silicon hydrogel CL brand was Biofinity (CooperVision, San Ramon, CA, USA), made of Comfilcon A (containing poly-2-hydroxyethylmethacrylate and silicone chains), swelling 48%.

### 2.5. Plasma Generation

Plasma was generated by the DC transient spark corona both in positive and negative arrangement, depicted on schemes in Figure 1. The working electrode was made of a medical intradermal needle and was adjustable with a micrometer screw; the grounding electrode was made of the platinum wire. The positive discharge burned at 9 kV and 300 μA in the regime of transient spark with a spark duration of 25–100 ns and a spark frequency of 800–1000 Hz; the given values varied in given intervals during the exposure and were not evaluated in detail. The negative discharge burned at 9 kV and 350 μA in the regime of pulseless glow corona. For details, see Khun et al. [17].

### 2.6. Experimental Design

All experiments with *Acanthamoeba* cysts were performed in a total volume of 1.5 mL per well in a 24-well cell culture plate (Corning Costar 3527) at room temperature, and each of them was performed in quintuplicate (in five parallels). In each experiment, control untreated cysts were included (one well per time interval and discharge type). Each set of five parallel experiments was performed in the same day.

### 2.7. Exposure of Acanthamoeba Cysts

#### 2.7.1. Cysts in Suspension

A total of 1.4 mL of sterile distilled water was pipetted into each experimental well, and 100 µL of properly re-suspended stock cyst suspension, i.e., 10^5^ cysts, was added into each well. The plate was covered with a perforated Teflon lid accurately printed on a 3D printer, which allows the passage of the working electrode (the diameter of the individual holes corresponded to the diameter of the electrode) and, at the same time, provides a closed experimental system. The cysts were exposed to either positive or negative discharge for 20, 25, 30, 35, 40, 50 and 55 min each. During the exposure, the temperature of the liquid did not exceed 40 °C.

#### 2.7.2. Cyst-Contaminated Contact Lenses

The CL was removed from the original package using tweezers (Dumont Style 7) and immersed into 1.4 mL of distilled water in a well so that the CL was placed with the concave (inner) side up. Immediately before use, the re-suspended cyst stock was 10× diluted with distilled water to a concentration of 10^5^ cysts·mL^−1^, and 100 µL of the suspension (10^4^ cysts) was pipetted on the lens’s concave surface. The plate was covered as above, allowed to stand for 30 min at ambient temperature, and then the cyst-contaminated lenses were exposed to negative discharge for 35, 40, 45, 50, 55 and 60 min.

### 2.8. Evaluation of Cysts Viability

The viability of *Acanthamoeba* cysts was assessed by spontaneous excystation, i.e., recovery of trophozoites from viable cysts in culture by inoculation of cysts (1) onto non-nutrient agar plates covered with heat-inactivated *E. coli* (a routine diagnostic cultivation method for determining *Acanthamoeba* infection; all experiments) and (2) into the liquid BCS medium (experiments with cysts in suspension) (for details, see below). Agar plates were prepared in wells of a 6-well culture plate (Corning Costar 3506; 3 mL of 1.5% non-nutrient agar per well covered with 30 µL of suspension of *E. coli*); the liquid BCS medium with antibiotics was dispensed into a 12-well culture plate (Corning Costar 3512) in a volume of 1.5 mL per well. The inoculated plates were incubated at 25 °C. The presence and growth of *Acanthamoeba* trophozoites were observed daily up to the seventh day of incubation using an inverted microscope. When needed, the cysts from the primary culture were sub-cultured to a new agar plate.

#### 2.8.1. Cysts in Suspension

Immediately after exposure, the cysts were inoculated onto an agar plate and into the BCS medium using exactly the same procedure. Briefly, cysts in the experimental and control well were re-suspended by aspiration up and down (ten times) using a micropipette set at 300 µL, and exactly 100 µL of this suspension was pipetted onto an agar plate into a well and into the BCS medium.

#### 2.8.2. Cyst-Contaminated Contact Lenses

After exposure, each experimental CL was carefully removed from the well with tweezers (the same type as above) and transferred to an agar plate so that its inner side containing the cysts faced the surface of the plate. To avoid contamination between wells, the tweezer was sterilized by immersion in absolute ethanol for 10 min after the transfer of each CL.

### 2.9. Physical Parameters of Contact Lenses

The following properties of the CLs were observed: the optical power, diameter, curvature and water content. Optical power was measured on a PL 2501 projection focus meter (Nikon, Minato, Tokyo, Japan) in immersion. CL diameter and curvature were measured on an Optimec CL optical analyzer (Lambda Polytech, Ilfracombe, UK). The water content was determined by weighing the swollen CL and the same CL dried to a constant weight at 80 °C. The observed parameters were compared for the CLs before and after plasma exposure for 30, 60 and 90 min. The measurements were repeated five times.

### 2.10. Infrared and Raman Spectroscopy

Infrared spectra in the range of 4000–400 cm^−1^ were recorded on the Nicolet 6700 FTIR spectrometer (Thermo Fisher Scientific, Waltham, MA, USA). Raman spectra were recorded on the DXR Microscope dispersion spectrometer (Thermo Fisher Scientific, Waltham, MA, USA). The recorded spectra were compared for the CLs before and after plasma exposure (in five parallels).

## 3. Results

### 3.1. Inactivation of Cysts

The results of cyst inactivation performed in aqueous suspension are summarized in Table 1. The cyst viability was assessed qualitatively by the presence/absence of viable trophozoites in culture using two cultivation methods, on an agar plate and in a liquid medium (see Materials and Methods), performed simultaneously in each experiment. For simplicity, the results obtained from both cultivation methods were evaluated together. Because each experiment was repeated five times in each time interval and discharge, the value in each table cell represents the result of ten determinations expressed as a percentage. Thus, a value of 10 means that cyst inactivation was demonstrated by only 1 of 10 cultures, i.e., 10% inactivation, and a value of 100 means complete inactivation, after which no trophozoite was detected in any of the 10 cultures, i.e., 100% inactivation of cysts. Complete inactivation occurred after 40 min of exposure to the negative discharge, while a positive discharge did not lead to complete inactivation even after 60 min.

As demonstrated above, the positive discharge proved to be insufficiently effective against *Acanthamoeba* cysts. Therefore, in the second series of experiments, the inactivation of cysts on the surface of CLs was performed only by negative discharge, and the viability of the cyst was evaluated only by cultivation on agar plates. Although the efficiency of both cultivation methods for excystation, i.e., for the recovery of trophozoites, is comparable, in experiments with contaminated CLs, we used only agar plates for easier handling of the CLs. Thus, the percentage of inactivation shown in Table 2 was determined from five cultures per time interval. Complete inactivation of *Acanthamoeba* cysts was achieved on the studied CL after 50 min of exposure.

### 3.2. Physical Parameters of Lenses

The optical power, diameter, curvature and water content were observed. No changes greater than 2% of the values measured before exposure were observed in any of the monitored parameters; the measured values are therefore not reported here in extenso.

### 3.3. Spectral Properties

The infrared spectra of Soflens and Biofinity CLs are shown in Figure 2. The curves show the superimposed spectra of the CLs before and after plasma exposure for 90 min. The course of the curves is only almost imperceptibly different, which shows that the spectral properties of the CLs are practically identical.

The same effect is evident in Figure 3, showing superimposed Raman spectra of the CLs before and after plasma exposure for 90 min.

## 4. Discussion

The presented results demonstrate the ability of plasma to inactivate *Acanthamoeba* in the stage of resistant cysts. Cyst stage was chosen for the experiments because it is this dormant stage to which acanthamoebae transform under adverse conditions and which is usually found in contaminated CL cases. Therefore, any disinfection systems for CLs should focus primarily on the inactivation of *Acanthamoeba* cysts, which are much more resistant than vegetative trophozoites, and, therefore, the vast majority of experimental works testing efficacy of disinfection solutions used cysts [8,9,53]. Other experiments, data not given, were performed also with various numbers of cysts. It can be stated that with increasing numbers of cysts, a longer time is required for inactivation. However, the selected numbers were chosen because they are significantly higher than the ordinary occurrence of amoebae on CLs.

In our experience, different microbes exhibited different sensitivity to NTP; while bacteria could be completely inactivated within seconds to minutes, yeasts required exposure for several minutes. Mold spores can survive exposures lasting several tens of minutes, comparable exposure times require sporulated bacteria [54] or microorganisms in the form of a biofilm, which are considerably more resistant to the microbicidal action of plasma in comparison with their planktonic forms [42]. Different species of fungi showed significant differences in the time of exposure needed to inactivate them, see reviews [44,55]. In addition, significant differences were observed in all of these studies between various modes of NTP production, namely between positive and negative DC transient spark corona discharges and between corona and dielectric barrier discharges [44].

Our results show that *Acanthamoeba* cysts are primarily sensitive to the negative discharge. The positive discharge did not inactivate the cysts even after one hour of exposure, which was the maximum time interval tested. However, the exact cause of the observed differences is unknown. In our previous works [55,56,57], we observed a rather better effectiveness of the positive discharge. In the case of amoebic cysts, the positive discharge, in contrast to the negative one, was not able to inactivate them even after an hour of exposure. However, these results are not fully comparable, as the former were obtained on bacteria or fungi and in an open exposure system.

Both negative and positive NTP produce hydrogen peroxide as the stable active compound [37]. Hydrogen peroxide is widely used in CL disinfection solutions. It has previously been shown by testing various CL solutions and disinfecting systems that 3% hydrogen peroxide-based systems kill *Acanthamoeba* cysts, i.e., they are cysticidal after 4–8 h depending on the rate of neutralization of the hydrogen peroxide [58]. Enhanced cysticidal effect of hydrogen peroxide was observed by the addition of certain peroxidases [59] or by catalysts with a delayed neutralizing effect [60]. Although hydrogen peroxide, together with nitrogen acids and various radical oxygen and nitrogen species (ROS and RNS), is generated by the discharge in PAW [37], it is difficult to speculate whether and to what extent the individual compounds/elements contribute to cysticidal activity. Moreover, for many reasons, including volatility and the half-life of ROS and RNS elements, the exact composition of the NTP in a particular system and/or time interval is not known.

Using negative discharge NTP, we achieved the complete inactivation of *Acanthamoeba* cysts after exposure for 40 and 50 min in suspension and on contaminated CLs, respectively. Exposure times correspond to or are slightly longer than those necessary to inactivate the resistant stages of other pathogenic microorganisms, e.g., mold or bacterial spores [54]. Biofilm-forming microorganisms must also be exposed to NTP for several minutes before they are completely inactivated [42]. In contrast, significantly shorter exposure times have been reported for non-sporulating bacteria [54], and also for *Acanthamoeba* cysts by [15], which to the best of our knowledge is the only publication to date on the plasma inactivation of *Acanthamoeba*. The cold atmospheric gas plasma (CAP) they used produced inactivation of *Acanthamoeba* cysts within 2–4 and 12 min on the surface of cellulose acetate filter and on CLs, respectively [15]. However, there are major differences between the experimental designs, including different plasma generation methods, of which the results are not comparable. In our experience, significant differences were observed between various modes of NTP production, namely between positive and negative DC transient spark corona discharges and between corona and dielectric barrier discharges [44].

Important findings for the possible practical use of plasma to disinfect CLs come from testing the sensitivity of the CLs themselves to the discharge. To the best of our knowledge, this is the first time that the physical parameters of CLs exposed to plasma were measured. Heaselgrave et al. [15] did not analyze the CLs themselves when monitoring the plasma efficacy on acanthamoebae on the surface of CLs. Observations of several physical parameters and infrared and Raman spectra of the commonly used hydrogel and silicon hydrogel CLs showed that the properties of the CLs are almost unaffected by long-term plasma exposure. The hard-to-see changes in the IR and Raman spectra can also be attributed to the different degrees of hydration (swelling) of the lenses´ material, but no changes in their chemical composition were detected.

Considering the possible practical use of activated plasma for the disinfection of CLs, previous findings regarding the effectiveness of NTP against CL-contaminating microbial pathogens, which can cause keratitis in CL wearers, e.g., *Pseudomonas aeruginosa*, *Staphylococcus aureus*, *Candida albicans* and *Aspergillus* spp. (from numerous reviews, see, e.g., [28]), should also be mentioned.

## 5. Conclusions

The non-thermal plasma produced by the negative DC transient spark corona discharge completely inactivates *Acanthamoeba* cysts within 40 min in water suspension and within 50 min on contact lenses. The plasma does not change the properties of exposed lenses; therefore, it can be used as an alternative method of disinfecting contact lenses.

## Figures and Tables

**Figure 1 microorganisms-09-01879-f001:**
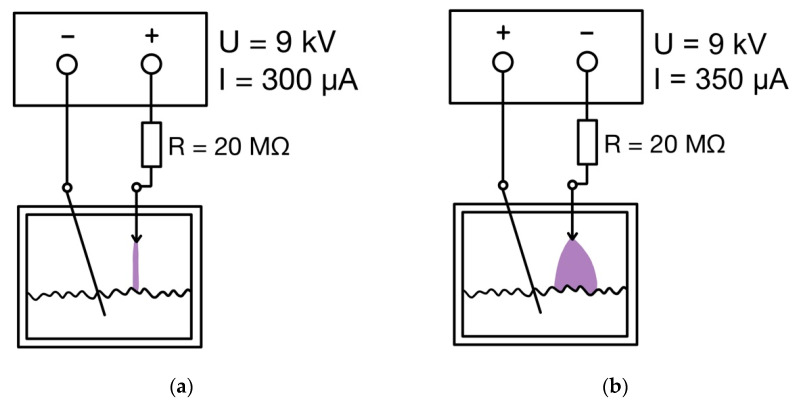
Wiring diagram of the apparatus for generating positive (**a**) and negative (**b**) discharge.

**Figure 2 microorganisms-09-01879-f002:**
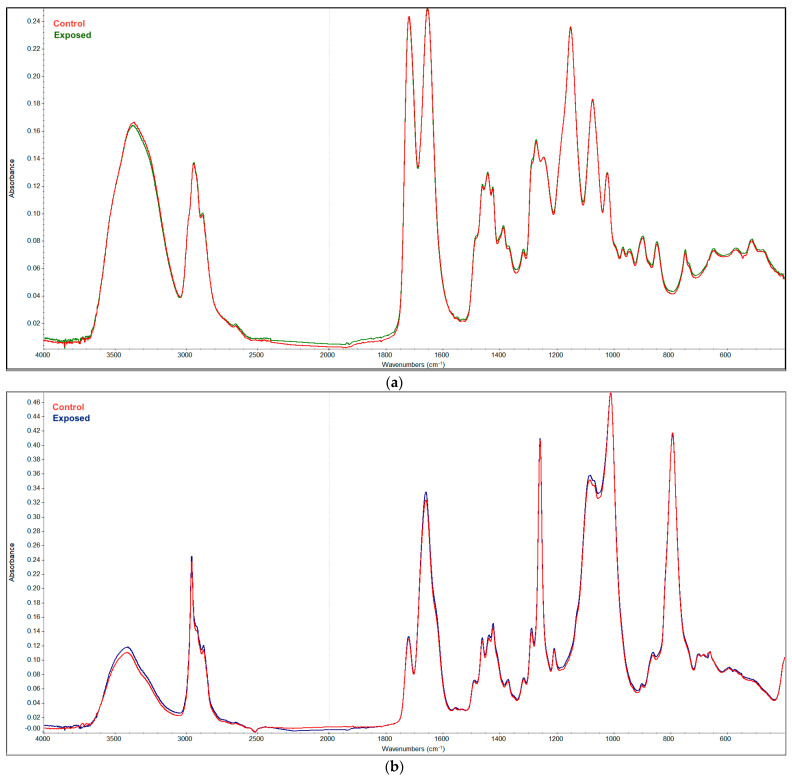
Cumulated FTIR spectra of Soflens (**a**) and Biofinity (**b**) CLs before and after plasma exposure for 90 min.

**Figure 3 microorganisms-09-01879-f003:**
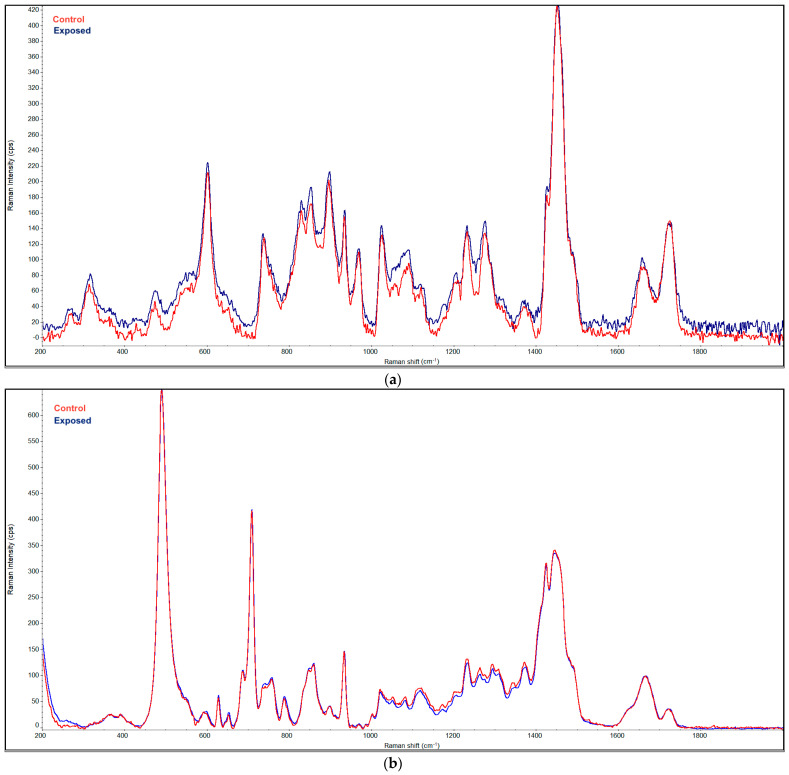
Cumulated Raman spectra of Soflens (**a**) and Biofinity (**b**) lenses before and after plasma exposure for 90 min.

**Table 1 microorganisms-09-01879-t001:** Efficacy of positive and negative discharge to inactivate *Acanthamoeba* cysts in suspension. Inactivation of cysts after discharge exposure is expressed as a percentage.

Exposure (Min)	Positive Discharge	Negative Discharge
20	0	10
25	0	20
30	20	60
35	10	80
40	60	100
45	80	100
50	80	100
55	60	100

**Table 2 microorganisms-09-01879-t002:** Efficacy of negative discharge to inactivate *Acanthamoeba* cysts on surface of contact lenses. Inactivation of cysts contaminating surfaces on Soflens (Bausch&Lomb) and Biofinity (CooperVision) contact lenses is expressed as a percentage.

Exposure (Min)	Soflens	Biofinity
35	60	20
40	80	40
45	80	80
50	100	100
55	100	100

## Data Availability

Data available on request due to restrictions e.g. privacy or ethical.

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
