# Peer review of "Inactivation of Acanthamoeba Cysts in Suspension and on Contaminated Contact Lenses Using Non-Thermal Plasma"

_microorganisms, 2021, doi:10.3390/microorganisms9091879_

Round 1
Reviewer 1 Report
Thanks for giving me an opportunity in reviewing this manuscript. Please see the attached document for the detailed report.

Reviewer 2 Report
This manuscript describes the use of non-thermal plasma generated by high voltage DC discharge as a method to sterilise contact lens inoculated with Acanthamoeba cysts. The paper is very well written and very clear. The claims that cysts were deactivated by the treatment were justified by the results and the fact that the lens were unaffected well documented by spectral analysis, however I wonder if the lenses were tested by a CL wearer before and after treatment (having been thoroughly washed to remove any remaining detrimental ions)?
Line 135. (2.5. Plasma generation). More details on the generation of the DC sparks would be helpful here. What is the duration of the transient? How much does the temperature of the liquid increase as a result of the passage of the spark? (was this measured?).
Line 139-140 current given as 300 and 350 A whereas in figure 1 currents are 300uA and 350uA. I suspect the uA are correct otherwise the energy dissipated would be huge! Also, uA units were mentioned in the cited Khun et al, 2018 paper.
Line 322. (5. Conclusions). Do the authors really envisage contact lens users having 5KV DC units in their homes?
Round 2
Reviewer 1 Report
Thanks for giving me an opportunity to review the manuscript. Authors have addressed my comments successfully.